REGISTERED REPORT PROTOCOL

# Gender-specific design and effectiveness of non-pharmacological interventions against cognitive decline and dementia–protocol for a systematic review and meta-analysis

**Andrea E. Zuelke**[1]*, **Steffi G. Riedel-Heller**[1], **Felix Wittmann**[1], **Alexander Pabst**[1], **Susanne Roehr**[1,2‡], **Melanie Luppa**[1‡]

**1** Institute of Social Medicine, Occupational Health and Public Health (ISAP), Medical Faculty, University of Leipzig, Leipzig, Germany, **2** Global Brain Health Institute (GBHI), Trinity College Dublin, Dublin, Ireland

‡ These authors are joint senior authors on this work.
* andrea.zuelke@medizin.uni-leipzig.de

## Abstract

### Introduction

Dementia is a public health priority with projected increases in the number of people living with dementia worldwide. Prevention constitutes a promising strategy to counter the dementia epidemic, and an increasing number of lifestyle interventions has been launched aiming at reducing risk of cognitive decline and dementia. Gender differences regarding various modifiable risk factors for dementia have been reported, however, evidence on gender-specific design and effectiveness of lifestyle trials is lacking. Therefore, we aim to systematically review evidence on gender-specific design and effectiveness of trials targeting cognitive decline and dementia.

### Methods and analysis

We will conduct a systematic review in accordance with the Preferred Reporting Items for Systematic Reviews and Meta-Analyses (PRISMA) guidelines. Databases MEDLINE (PubMed interface), PsycINFO, Web of Science Core Collection, Cochrane Central Register of Controlled Trials (CENTRAL) and ALOIS will be searched for eligible studies using a predefined strategy, complemented by searches in clinical trials registers and Google for grey literature. Studies assessing cognitive function (overall measure or specific subdomains) as outcome in dementia-free adults will be included, with analyses stratified by level of cognitive functioning at baseline: a) cognitively healthy b) subjective cognitive decline 3) mild cognitive impairment. Two reviewers will independently evaluate eligible studies, extract data and determine methodological quality using the Scottish Intercollegiate Guidelines Network (SIGN)-criteria. If sufficient data with regards to quality and quantity are available, a meta-analysis will be conducted.

**Data Availability Statement:** All relevant data from this study will be made available upon study completion.

**Funding:** This work was supported by the German Federal Ministry of Health (BMG; url: https://www.bundesgesundheitsministerium.de/), grant number: ZMVI1-2520FSB510. The grant was awarded to SR-H and SR. The funders had and will not have a role in study design, data collection and analysis, decision to publish, or preparation of the manuscript. We acknowledge support from Leipzig University for Open Access Publishing.

**Competing interests:** The authors have declared that no competing interests exist.

## Ethics and dissemination

No ethical approval will be required as no primary data will be collected.

## PROSPERO registration number

CRD42021235281.

## Introduction

Currently, over 50 million people are living with dementia worldwide, with projected increases to over 150 million in 2050 [1]. Dementia inflicts a tremendous burden on individuals and their relatives who often act as caregivers [2]. In 2015, the World Health Organization (WHO) estimated the global costs of dementia to be 818 billion US-dollars, with approximately 85% of the costs being related to family- and social care. Moreover, dementia is among the leading causes of disability in older age, accounting for 11.9% of years lived with disability due to non-communicable diseases, making it a public health priority [3, 4].

In the absence of curative treatment, the pivotal role of identifying modifiable risk factors and designing preventive strategies for dementia has been highlighted, and knowledge on such factors is evolving rapidly [4, 5]. To date, there is evidence for a variety of potentially modifiable factors increasing risk for dementia: low education in early life, hearing loss, traumatic brain injury, hypertension, obesity, excessive alcohol consumption (>21 units per week) in midlife, diabetes mellitus, depression, physical inactivity, smoking, social isolation, and exposure to air pollution in later life. It is estimated that these risk factors hold responsible for approximately 40% of dementia cases in high-income countries (HIC), with preventive potential in low- and middle-income countries (LMIC) being even higher [6]. While the first generation of intervention studies mainly targeted single such risk factors, more recent trials focused on multi-domain interventions, accounting for the multifactorial etiology of dementia [5, 7]. Several multi-domain interventions, first and foremost the FINGER-study (Finnish Geriatric Intervention Study to Prevent Cognitive Impairment and Disability; [8]), have been completed, exhibiting positive effects on cognitive outcomes in subpopulations with increased dementia risk [9–11].

Several gender differences regarding prevalence and presentation of dementia as well as risk factors have been reported. Women's risk for Alzheimer's disease (AD), the most common type of dementia (up to 80% of all cases), is twice as high as men's [12]. This is in part due to sex differences in life-expectancy, with age being the single most important risk factor for dementia [13]. On the other hand, while known risk factors for stroke and vascular dementia such as atherosclerosis, diabetes, obesity, hypertension, atrial fibrillation and heart failure are more common in men, evidence suggests that these risk factors disproportionately increase women's risk for dementia [14, 15]. Depression constitutes another important risk factor for cognitive decline and dementia which is unequally distributed between genders (prevalence women vs. men = 2:1; [16]), with studies reporting up to 70% increased risk for AD in midlife depression [17]. Further, men and women reporting feelings of loneliness are at increased risk for developing dementia. However, while certain studies found stronger associations between loneliness and dementia onset in men [18], others reported no gender differences [19]. Social networks and social participation have further been linked to dementia risk and cognitive functioning in older age: In a large systematic review, social activity and social network size were linked to better cognitive function in late life, irrespective of gender [20]. Widowhood in

older age is more common in women since they tend to outlive their husbands [6]. Losing one's spouse can affect cognitive function, due to changes in social networks and possibly increased depressive symptoms; however, investigations on potential gender differences in the association between spousal loss and cognitive function revealed mixed results [21, 22]. Furthermore, gender differences regarding physical activity, motivation and uptake of being physically active have been reported [23–25]. While low to moderate alcohol consumption is associated with lower dementia risk [26], excessive or at-risk-drinking has been identified as a risk factor for dementia [27]. At-risk drinking has been reported more frequently in males than females across several countries in middle-aged and older adults [27, 28]. Prevalence of smoking has been reported higher in men than in women among middle-aged and elderly populations across Europe [29], with an inverse association between smoking prevalence and level of education observed in both genders [30]. On the other hand, rates of smoking cessation and long-term abstinence from tobacco were found to be lower in female than in male (ex-)smokers [29, 31].

Despite established gender differences regarding risk factors for cognitive decline and dementia, little is known about the consideration of gender in lifestyle trials. To the best of our knowledge, no systematic review has yet addressed this question. Adequately addressing potential gender differences could support the development of tailored gender-specific interventions against cognitive decline and dementia and increase their effectiveness.

### Objectives

Our review aims to systematically assess gender-specific design and effectiveness of lifestyle trials to reduce the risk of cognitive decline and dementia. Depending on the number and quality of eligible trials identified through systematic literature search, an additional meta-analysis will be conducted. This protocol outlines the objectives and strategy for the review, following the Preferred Reporting Items for Systematic Reviews and Meta-Analyses (PRISMA) for systematic review protocols (PRISMA-P) guidelines [32].

## Methods and analysis

Our review will be conducted in accordance with the Preferred Reporting Items for Systematic Reviews and Meta-Analyses (PRISMA) guidelines [33]. The planned process of literature search is outlined in **Fig 1**. This section describes the process of literature search, selection of studies and management of data.

### Eligibility criteria

We will include randomized controlled trials (RCTs) testing lifestyle interventions against cognitive decline and dementia, targeting adults with either unimpaired cognitive function, subjective cognitive decline (SCD) or mild cognitive impairment (MCI) at baseline. Interventions will need to be targeted at age-related cognitive decline. To be included in the review, studies need to assess one or more outcomes covering cognitive function using validated, standardized instruments at baseline and follow-up. Potential outcomes include either global assessments of cognitive function, as assessed e.g. with the Mini-Mental State Examination (MMSE; [34]) or the Montreal Cognitive Assessment (MoCA; [35]), or specific cognitive domains, e.g. memory, language or executive function.

Eligible RCTs apply single- or multi-domain interventions, including one of the following elements or combinations thereof: diet, physical activity, social activity, cognitive activity, depression, hearing ability, management of cardiovascular diseases or diabetes, smoking, alcohol consumption, and/or psychoeducation. No restriction will be made regarding mode of

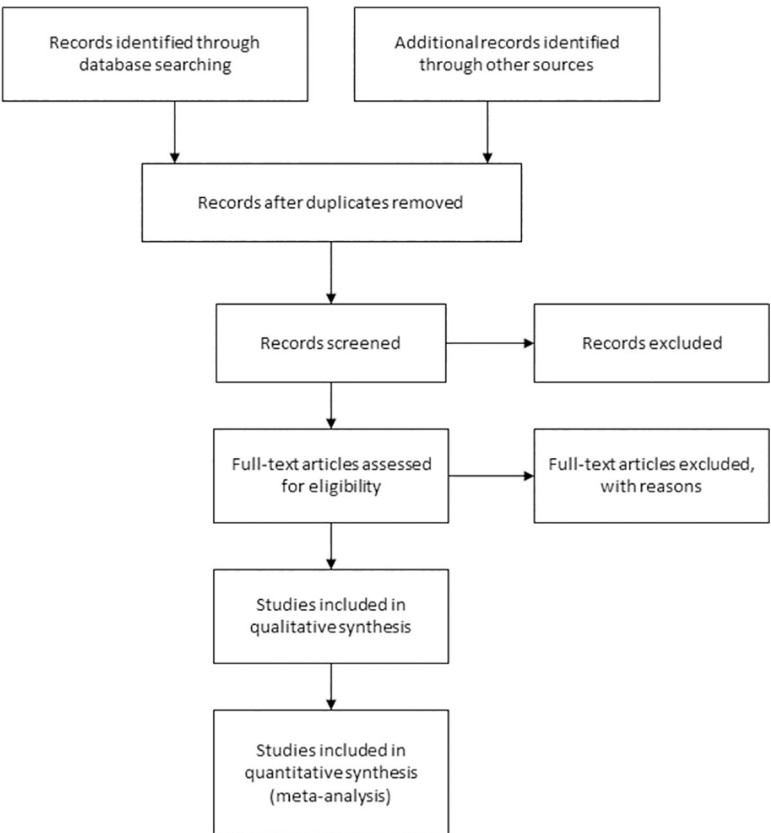

**Fig 1. Flow diagram of the planned study selection process adapted from the Preferred Reporting Items for Systematic Reviews and Meta-Analysis (PRISMA) statement.**

delivery of the intervention, e.g. individual face-to-face administration, internet- or mobile-based interventions or group settings. Articles written in English or German and published in peer-reviewed journals will be considered for inclusion in the systematic review, without restrictions regarding date of publication. We will include RCTs testing interventions against any kind of control condition, e.g. treatment as usual, passive, active or waitlist control.

We will exclude RCTs targeting individuals diagnosed with dementia or focusing on samples with severe pre-existing conditions, e.g. myocardial infarction, cancer, stroke, or psychiatric diseases, e.g. major depression, as these conditions are linked to impaired cognitive function and increasedd risk for cognitive decline and dementia [36]. Further, RCTs addressing post-operative cognitive function will not be eligible for the review. RCTs focusing exclusively on pharmacological interventions or using non-standardized assessments of cognitive function will not be eligible for the review. Only original studies will be included, excluding dissertations, book chapters, editorials or brief communications.

Our review aims to describe how and to what extent identified RCTs address gender in their design and implementation, thereby providing an overview of the current state-of-the-art and pointing out possibilities for optimization with regards to gender-specificity. Since most RCTs conducted to date apply a dichotomous assessment of male or female sex, consideration of a broader spectrum of gender identities in our review might be limited. We will, however, include evidence from populations with non-binary gender identities in our review if reported in identified RCTs. Possible implications of gender roles and attributes will be adequately

addressed when interpreting the results. Only RCTs reporting gender-specific results will be considered eligible for the systematic review.

## Search strategy

Literature searches will be conducted in MEDLINE (PubMed interface), Cochrane Central Register of Controlled Trials (CENTRAL), PsycINFO, Web of Science (Science Citation Index Expanded, Social Sciences Citation Index) and ALOIS, a specialized register of studies on dementia and cognitive decline, maintained by the Cochrane Dementia and Cognitive Improvement Group. Additional searches will be conducted using Google for grey literature. Moreover, registers of clinical trials (ClinicalTrials.gov, EU Clinical Trials Register, WHO International Clinical Trials Registry Platform Search Portal, German Clinical Trials Register) will be screened. Finally, we will scan reference lists of recent reviews and meta-analyses on lifestyle interventions against cognitive decline and dementia and reference lists of included trials to identify potentially missed studies.

A combination of a wide range of search terms will be used to build the search string, using combinations of MeSH-terms or other controlled vocabulary (where possible) and key words. The draft for the search strategy to be applied in the respective databases is included in **S1 Appendix**. The process of the literature search and study selection will be summarized using the PRISMA flow diagram.

## Data management

We will use Review Manager (RevMan) software package, version 5.4 (provided by The Cochrane Collaboration, 2020), for management of references and data. If sufficient data can be retrieved, meta-analysis will be conducted using Stata 16.0 (SE; StataCorp LP, College Station, Texas, USA).

## Selection process

Two reviewers (AZ and FW) will independently screen titles and abstracts of studies identified through literature search to select studies potentially suitable for the systematic review, according to inclusion- and exclusion criteria described above. In case of uncertainty, the respective study will be read in full text by the two review authors. Disagreement regarding suitability of potentially relevant studies will be resolved by consulting a third review author (ML or SR-H). Reasons for exclusion of articles will be provided.

## Data collection

Data extraction will be handled independently by two review authors (AZ and FW), using a standardized, pre-piloted extraction form. Reliability of data collection will be assessed in a random sample of included studies. We will extract the following data:

1. Study description, e. g. first author, year of publication, country

2. Aim of study

3. Type of intervention

4. Sample description, e. g. sample size of intervention- and control group(s), recruitment strategy, intervention design/type, control group, outcomes assessed, type of assessment, inclusion/exclusion criteria, duration of intervention, length of follow-up

5. Sample characteristics: e. g. mean age, age range, gender distribution

6. Methods of data collection

7. Information on dropout, handling of missing data, imputation techniques applied (if applicable)

8. Information on type of data analysis, i.e. intention-to-treat- or completer-analyses

9. Methodological aspects: risk of bias, study limitations

Potential discrepancies between reviewers will be resolved through discussion with a third author (ML or SR-H). Missing data will be requested from study authors. If possible, we will extract data from intention-to-treat-analyses, reflecting the initial group assignment of participants. In cases where intention-to-treat-data are unavailable, we will extract results of per-protocol-analyses. Our main outcome will be the standardized mean difference between intervention- and control group at post intervention. If sufficient data from post-intervention follow-up is available, we will assess stability of effects. In cases of multiple follow-up assessments, we will use data from the assessment closer to the mean time interval of follow-up periods.

## Quality assessment

Methodological quality and risk of bias of included studies will be assessed by two review authors (AZ and FW) independently, using the SIGN (Scottish Intercollegiate Guidelines Network)-criteria for RCTs [37]. This set of criteria includes information on randomization, allocation concealment, blinding of participants and outcome assessors, level of drop-out before study completion and overall-criteria for rating study quality, e.g. directness and certainty of intervention effects. Quality of included studies will be assessed parallel to the process of data extraction. Disagreement between the two review authors will be resolved by discussion, with involvement of a third review author (ML or SR-H) if no consensus can be reached. If necessary, study authors will be contacted to obtain any additional information needed to assess methodological quality of studies.

## Data synthesis and presentation

Results of included studies will be reported in narrative synthesis and tables, displaying 1) study characteristics, 2) sample characteristics and 3) overall-results. For each included study, we will provide numbers of male and female participants, gender-specific covariates and gender-specific effects of interventions. Beyond that, we will provide a narrative synthesis of gender aspects of included studies, e.g. consideration of gender in the study design, presentation of results, discussion, conclusion or recommendations, if feasible. If sufficient data is available in terms of quantity and quality, meta-analysis will be conducted, including forest plots and estimation of a pooled overall effect size. Analyses will be conducted separately for men and women, respectively. Funnel plots and $I^2$-statistics will be applied to assess levels of heterogeneity and Egger's tests will be applied to assess potential bias caused by small sample sizes. Depending on the level of observed heterogeneity, random-, fixed- or mixed-effects meta-analyses will be conducted. Effectiveness of interventions will be assessed using standardized mean group differences, particularly Hedge's *g* as effect size. We will conduct meta-regressions with a dummy variable for gender in order to assess differences in effectiveness between men and women. To account for possible effects of age as the single most important risk factor for dementia, we will assess both unadjusted and age-adjusted treatment effects. Narrative synthesis and meta-analyses will be conducted separately for different levels of cognitive function, i.e. 1) cognitively healthy individuals, 2) SCD, 3) MCI.

## Discussion

As dementia currently cannot be cured, risk reduction strategies have been highlighted as essential in lowering the burden of disease to counteract the projected dementia epidemic [5]. The planned systematic review will for the first time provide evidence on how gender differences are considered in the design and effectiveness of lifestyle interventions against cognitive decline and dementia. Thereby, we will point out whether potential gender differences are already being taken into account in the design and execution of dementia prevention efforts or whether possibilities for improvement prevail. Knowledge on whether existing intervention efforts have differential effects depending on gender is crucial for the design of future prevention strategies and personalized interventions to preserve cognitive function. Systematically assessing aspects of gender in existing trials could help facilitate the design of future interventions aimed at preserveing cognitive function. By addressing these questions, the results of our review will contribute valuable knowledge on which types of interventions work for men and women, respectively, to prevent risk of cognitive decline and dementia. Adequately taking into account gender differences in dementia risk factors might lead to more targeted efforts and improve the effectiveness of risk reduction strategies against cognitive decline and dementia. This might in turn contribute to preserve cognitive function in ageing populations, and it even has the potential to reduce prevalence of dementia.

### Dissemination plan

The results of the review will be published in an international peer-reviewed journal. Further, results will be presented at professional conferences and meetings. Extracted data will be available as supporting information upon completion of the review. The registry in PROSPERO will be updated regularly in the review process.

### Amendments

Any type of amendment to the original protocol will be documented, including date of, description of and reason for amendment.

## Supporting information

**S1 Checklist.**
(DOCX)

**S1 Appendix. Search strategy.**
(DOCX)

## Author Contributions

**Conceptualization:** Steffi G. Riedel-Heller, Susanne Roehr, Melanie Luppa.

**Data curation:** Andrea E. Zuelke, Felix Wittmann.

**Funding acquisition:** Steffi G. Riedel-Heller, Susanne Roehr.

**Investigation:** Andrea E. Zuelke, Felix Wittmann.

**Methodology:** Andrea E. Zuelke, Alexander Pabst, Melanie Luppa.

**Project administration:** Andrea E. Zuelke, Steffi G. Riedel-Heller, Melanie Luppa.

**Supervision:** Steffi G. Riedel-Heller, Melanie Luppa.

**Visualization:** Andrea E. Zuelke, Felix Wittmann.

**Writing – original draft:** Andrea E. Zuelke.

**Writing – review & editing:** Andrea E. Zuelke, Steffi G. Riedel-Heller, Felix Wittmann, Alexander Pabst, Susanne Roehr, Melanie Luppa.

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
