## [Decision Letter · Decision Letter 0]

10 Aug 2021

PONE-D-21-08933

Gender-specific design and effectiveness of non-pharmacological interventions against cognitive decline and dementia – Protocol for a systematic review and meta-analysis.

PLOS ONE

Dear Dr. Zuelke,

Thank you for submitting your manuscript to PLOS ONE. After careful consideration, we feel that it has merit but does not fully meet PLOS ONE’s publication criteria as it currently stands. Therefore, we invite you to submit a revised version of the manuscript that addresses the points raised during the review process.

Please ensure that all text references to PRISMA guidelines reflect that these are guidelines for systematic review reporting, not systematic review conduct.

We look forward to receiving your revised manuscript.

Kind regards,

Lisa Susan Wieland

Academic Editor

PLOS ONE

Journal Requirements:

5. Please note that in order to use the direct billing option the corresponding author must be affiliated with the chosen institute. Please either amend your manuscript to change the affiliation or corresponding author, or email us at plosone@plos.org with a request to remove this option.

6. Please upload a new copy of Figure 1 as the detail is not clear. Please follow the link for more information: https://blogs.plos.org/plos/2019/06/looking-good-tips-for-creating-your-plos-figures-graphics/" https://blogs.plos.org/plos/2019/06/looking-good-tips-for-creating-your-plos-figures-graphics/.

Reviewers' comments:

Reviewer's Responses to Questions

**Comments to the Author**

1. Does the manuscript provide a valid rationale for the proposed study, with clearly identified and justified research questions?

Reviewer #1: Yes

Reviewer #2: Yes

2. Is the protocol technically sound and planned in a manner that will lead to a meaningful outcome and allow testing the stated hypotheses?

Reviewer #1: Partly

Reviewer #2: Yes

3. Is the methodology feasible and described in sufficient detail to allow the work to be replicable?

Reviewer #1: Yes

Reviewer #2: Yes

4. Have the authors described where all data underlying the findings will be made available when the study is complete?

Reviewer #1: Yes

Reviewer #2: Yes

5. Is the manuscript presented in an intelligible fashion and written in standard English?

Reviewer #1: Yes

Reviewer #2: Yes

6. Review Comments to the Author

You may also provide optional suggestions and comments to authors that they might find helpful in planning their study.

Reviewer #1: I would like to thank the authors for submitting this protocol on a topic of high interest in the field. The protocol is well written, although it wasn't very clear to me what sex- or gender-specific information the authors are trying to extract from the trials, and how would these information help to improve future interventional study design? What if these information were not reported, will additional effort be made to obtain these data?

Reviewer #2: It is a very well written project with a description of the appropriate methodology to meet the proposed objectives.

7. PLOS authors have the option to publish the peer review history of their article (what does this mean?). If published, this will include your full peer review and any attached files.

Reviewer #1: No

Reviewer #2: No

---

## [Author Response · Author response to Decision Letter 0]

12 Aug 2021

Reviewer 1

„I would like to thank the authors for submitting this protocol on a topic of high interest in the field. The protocol is well written, although it wasn't very clear to me what sex- or gender-specific information the authors are trying to extract from the trials, and how would these information help to improve future interventional study design? What if these information were not reported, will additional effort be made to obtain these data?”

We thank the reviewer for their kind appraisal of our manuscript! The revised manuscript includes additional information on gender-specific information to be reported in the planned systematic review. Please see page 10, lines 242ff:

“For each included study, we will provide numbers of male and female participants, gender-specific covariates and gender-specific effects of interventions. Beyond that, we will provide a narrative synthesis of gender aspects of included studies, e.g. consideration of gender in the study design, presentation of results, discussion, conclusion or recommendations, if feasible.”

Lines 264ff: “Thereby, we will point out whether potential gender differences are already being taken into account in the design and execution of dementia prevention efforts or whether possibilities for improvement prevail. Knowledge on whether existing intervention efforts have differential effects depending on gender is crucial for the design of future prevention strategies and personalized interventions to preserve cognitive function. Systematically assessing aspects of gender in existing trials could help facilitate the design of future interventions aimed at preserveing cognitive function. By addressing these questions, the results of our review will contribute valuable knowledge on which types of interventions work for men and women, respectively, to prevent risk of cognitive decline and dementia.”

Reviewer 2

„It is a very well written project with a description of the appropriate methodology to meet the proposed objectives.”

We thank the reviewer for their kind appraisal of our manuscript! 

In addition to the requested changes, we updated the information on inclusion and exclusion criteria to be applied in the planned review. In amendment to the original protocol, we will exclude studies focusing on samples with severe pre-existing conditions, e.g. myocardial infarction, cancer, coronary heart disease, history of traumatic brain injury, stroke or psychiatric diseases, e.g. major depressive disorder, as these conditions are linked to impaired cognitive function and increased risk for cognitive decline and dementia. Our systematic review and meta-analyses focusses on the prevention of age-related cognitive decline (see page 6, line 142), whereas many RCTs focusing on subjects with severe pre-existing conditions aim at rehabilitation of cognitive function. We are confident that excluding respective study samples will result in a more clearly defined samples selection.

Please see page 6, lines 139: DELETED: “Our review will include both RCTs investigating healthy older adults and those at-risk for dementia due to underlying medical conditions, e.g. stroke, diabetes or coronary heart disease (not limited to these conditions).”

CHANGES: Page 7, lines 157ff: “We will exclude RCTs targeting individuals diagnosed with dementia or focusing on samples with severe pre-existing conditions, e.g. myocardial infarction, cancer, stroke, or psychiatric diseases, e.g. major depression, as these conditions are linked to impaired cognitive function and increasedd risk for cognitive decline and dementia.”

---

## [Decision Letter · Decision Letter 1]

17 Aug 2021

Gender-specific design and effectiveness of non-pharmacological interventions against cognitive decline and dementia – Protocol for a systematic review and meta-analysis.

PONE-D-21-08933R1

Dear Dr. Zuelke,

We’re pleased to inform you that your manuscript has been judged scientifically suitable for publication and will be formally accepted for publication once it meets all outstanding technical requirements.

Kind regards,

Lisa Susan Wieland

Academic Editor

PLOS ONE

Additional Editor Comments (optional):

Reviewers' comments:

Reviewer's Responses to Questions

**Comments to the Author**

1. Does the manuscript provide a valid rationale for the proposed study, with clearly identified and justified research questions?

Reviewer #1: Yes

2. Is the protocol technically sound and planned in a manner that will lead to a meaningful outcome and allow testing the stated hypotheses?

Reviewer #1: Yes

3. Is the methodology feasible and described in sufficient detail to allow the work to be replicable?

Reviewer #1: Yes

4. Have the authors described where all data underlying the findings will be made available when the study is complete?

Reviewer #1: Yes

5. Is the manuscript presented in an intelligible fashion and written in standard English?

Reviewer #1: Yes

6. Review Comments to the Author

You may also provide optional suggestions and comments to authors that they might find helpful in planning their study.

Reviewer #1: Thanks the authors for the revised version of the protocol, it is a well-written protocol for a potentially impactful study. I look forward to the results.

7. PLOS authors have the option to publish the peer review history of their article (what does this mean?). If published, this will include your full peer review and any attached files.

Reviewer #1: No

---

## [Editor Report · Acceptance letter]

20 Aug 2021

PONE-D-21-08933R1 

Gender-specific design and effectiveness of non-pharmacological interventions against cognitive decline and dementia – Protocol for a systematic review and meta-analysis. 

Dear Dr. Zuelke:

I'm pleased to inform you that your manuscript has been deemed suitable for publication in PLOS ONE. Congratulations! Your manuscript is now with our production department. 

Kind regards, 

on behalf of

Dr. Lisa Susan Wieland 

Academic Editor

PLOS ONE